# 3D Surface Reconstruction in the Wild by Deforming Shape Priors from Synthetic Data

## Abstract

We present a new method for category-specific 3D reconstruction from a single image. A limitation of current deep learning color image-based 3D reconstruction models is that they do not generalize across datasets, due to domain shift. In contrast, we show that one can learn to reconstruct objects across datasets through shape priors learned from synthetic 3D data and a point cloud pose canonicalization method. Given a single depth image at test time, we first place this partial point cloud in a canonical pose. Then, we use a neural deformation field in the canonical coordinate frame to reconstruct the 3D surface of the object. Finally, we jointly optimize object pose and 3D shape to fit the partial depth observation. Our approach achieves state-of-the-art reconstruction performance across several real-world datasets, even when trained without ground truth camera poses (which are required by some of the state-of-the-art methods). We further show that our method generalizes to different input modalities, from dense depth images to sparse and noisy LIDAR scans.

## 1 Introduction

Reconstructing 3D object surfaces from images is a longstanding problem in the computer vision community, with applications to robotics (Bylow et al., 2013) or content creation (Huang et al., 2017). Every computational approach aimed at 3D reconstruction has to answer the question of which representation is best suited for the underlying 3D structure. An increasingly popular answer is to use neural fields (Park et al., 2019; Mescheder et al., 2019) for this task. These neural fields, trained on 3D ground truth data, represent the de-facto gold standard regarding reconstruction quality. However, the reliance on 3D ground truth has, for now, limited these approaches to synthetic data. To remove the reliance on 3D data, the community has shifted to dense (Mildenhall et al., 2020), or sparse (Zhang et al., 2021) multi-view supervision with known camera poses. Similarly, single-view 3D reconstruction methods have also made considerable progress by using neural fields as their shape representation (Lin et al., 2020; Duggal & Pathak, 2022). While these single-view methods can be trained from unconstrained image collections, they have not achieved the high quality of multi-view or 3D ground truth supervised models. In this work, we aim to answer the question: *How can we achieve the reconstruction quality of 3D supervised methods from single view observations in the wild?*

With recent advances in generative modeling of synthetic 3D data (Gao et al., 2022), using 3D data for supervision has become practical once again. However, the problem of aligning image observations to canonical spaces remains challenging. One way to solve this alignment problem is to learn the camera pose from data (Ye et al., 2021). However, learning camera pose prediction from color images is a complex problem, and existing methods do not generalize to new datasets due to domain shifts. Another promising research direction is the use of equivariant neural networks. For example, Condor (Sajnani et al., 2022) and Equi-Pose (Li et al., 2021) use equivariant network layers to canonicalize complete and partial point clouds through a self-supervised reconstruction loss.

Given an image taken from a calibrated camera, instead of using ground truth camera poses during inference, as other single-view 3D reconstruction methods Lin et al. (2020); Duggal & Pathak (2022), we suggest using a single depth image together with a pretrained canonicalization network to register the partial point clouds to the canonical coordinate space. However, during our work we found that canonical reconstruction methods are extremely sensitive to deviations in the estimated

canonical pose (Section 4.5). To recover from bad registration results, we jointly fine-tune 3D shape and pose using only the partial shape (Figure 2). We achieve 3D reconstruction results on synthetic data close to or better than the state-of-the-art. Furthermore, we show that using depth images as input allows for generalization across various datasets, from dense depth in synthetic and natural images to sparse depth inputs from LIDAR scans.

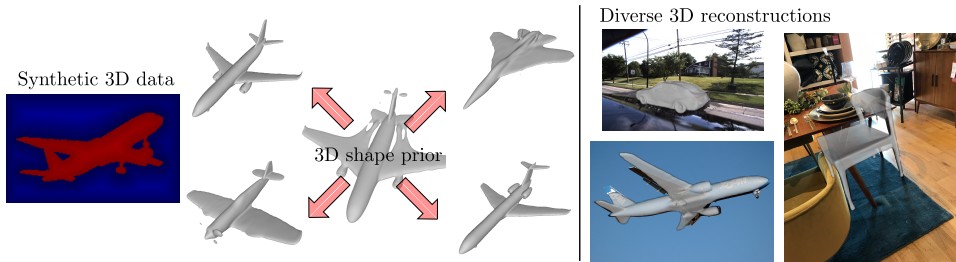

Figure 1: We leverage synthetic 3D data to learn a shape prior. Using a pose registration algorithm, we canonicalize partial point clouds to the canonical coordinate frame to generate diverse 3D reconstructions.

## 2    RELATED WORK

3D object reconstruction based on a conditional input, such as images or depth is an active research area (Chen et al., 2019; Mescheder et al., 2019; Mildenhall et al., 2020; Sitzmann et al., 2019; Tulsiani et al., 2017; Häni et al., 2020). The defacto gold standard in terms of reconstruction quality uses 3D ground truth data (Park et al., 2019; Mescheder et al., 2019). However, these approaches are largely limited to synthetic data, such as Shapenet (Chang et al., 2015). Reconstruction of real-world shapes has been performed by transferring the learned representation across domains (Duggal et al., 2022; Bechtold et al., 2021) or with the use of special depth sensors (Newcombe et al., 2011; Choe et al., 2021). However, collecting 3D ground truth data in the real world can be difficult. With the development of neural rendering and inverse graphics methods, the requirement for 3D ground truth has been relaxed in favor of dense multi-view supervision (Xu et al., 2019; Mildenhall et al., 2020; Goel et al., 2022; Zhang et al., 2021) or single view methods that require ground truth camera poses (Lin et al., 2020; Duggal & Pathak, 2022). However, not all applications allow for the collection of multi-view images, and estimating camera poses from images remains challenging. With the advent of generative models for 3D shapes (Gao et al., 2022), using 3D supervision has become an interesting prospect once more. However, these 3D models are all living in a canonical coordinate frame. Our work shows how we can leverage such canonical 3D data for shape reconstruction in the wild.

### 2.1    3D RECONSTRUCTION FROM SINGLE VIEWS

There have been extensive studies on 3D reconstruction from single view images using various 3D representations, such as voxels (Yan et al., 2016; Tulsiani et al., 2017; Wu et al., 2018; Yang et al., 2018; Wu et al., 2018; 2017), points (Fan et al., 2017; Yang et al., 2019), primitives (Deng et al., 2020; Chen et al., 2020) or meshes (Kanazawa et al., 2018; Goel et al., 2022). Most of the methods above use explicit representations, which suffer from limited resolution or fixed topology. Neural rendering and neural fields provide an alternative representation to overcome these limitations. Recent methods showed how to learn Signed Distance Functions (SDFs) (Xu et al., 2019; Lin et al., 2020; Duggal & Pathak, 2022) or volumetric representations such as occupancy (Ye et al., 2021), which have shown great promise in learning category-specific 3D reconstructions from unstructured image collections. However, these methods usually require additional information, such as ground truth camera poses, which limits their applicability. In our work, we propose a method that does not require ground truth camera poses and leverages widely available synthetic data to learn a category-specific 3D prior model.

## 2.2 Learning Shape Reconstruction through Deformation

Learning a generalizable model that maps a low-dimensional latent code to 3D surfaces can suffer from low-quality reconstructions. Category-specific deformable shape priors are useful to improve the quality of the reconstruction (Blanz & Vetter, 1999; Engelmann et al., 2017; Kanazawa et al., 2018; Kar et al., 2015; Loper et al., 2015; Mitchell et al., 2019). These methods generally learn the deformation to an initial base shape. More recent work has used neural rendering together with SDFs (Lin et al., 2020; Duggal & Pathak, 2022) to learn 3D shape priors from image collections and their associated camera poses. Other methods (Deng et al., 2021; Zheng et al., 2021) jointly learn the deformation and the template shape in a canonical frame. In this work, we go one step further and show how we can leverage template shape and deformation models for incomplete observations registered to the template coordinate frame.

## 2.3 Pose Registration and 3D Shape Canonicalization

Reliance on camera poses is an issue for many real-world datasets but a necessary step for neural rendering or deformation-based models. Point cloud registration can estimate the object pose directly and has achieved good performance when matching point clouds of the same object; however, these methods are unsuitable for single view pose estimation without a ground truth 3D model (Jiang et al., 2021; Wu et al., 2021; Qin et al., 2022). Category-level object pose estimation methods achieve tremendous results, for supervised training mechanisms (Rempe et al., 2020; Novotny et al., 2019; Wang et al., 2019), and using only self-supervision (Spezialetti et al., 2020; Sun et al., 2021; Li et al., 2021; Sajnani et al., 2022; Katzir et al., 2022). For example, Canonical Capsules (Sun et al., 2021) learn to represent object parts with pose-invariant capsules by training a Siamese network in a self-supervised manner. Although the learned capsules can reconstruct the input point cloud in the learned canonical frame, Canonical Capsules only works on complete point clouds. In contrast, Equi-pose (Li et al., 2021) can canonicalize both complete and partial point clouds. By leveraging an SE(3) equivariant network, Equi-pose simultaneously learns to estimate object pose and canonical point cloud completion. Our work shows that one can leverage Equi-pose with test time pose refinement to get accurate shape reconstructions in canonical space.

## 2.4 Pointcloud Completion

Instead of relying on ground truth camera poses, we use depth images to register the partial 3D point cloud into a canonical frame. As we use depth images as input, our method closely relates to point cloud completion algorithms. Early work used 3D convolutions to learn shape completion (Dai et al., 2017; Huang et al., 2020). However, 3D convolutions are costly and operate on a canonical voxel grid. More recently, PointNet encoders were used for shape completion (Liu et al., 2020; Yuan et al., 2018). Transformers have also been shown to work well on this task (Yu et al., 2021; Yan et al., 2022). However, these methods rely on points already in a canonical coordinate frame. Further, these methods do not reconstruct the underlying surface of the object but output a limited number of points. In contrast, out method does not rely on canonical input points and reconstructs the underlying object surface with high fidelity.

# 3 Method

Given a single segmented RGB-D image of an object, our goal is to reconstruct the underlying 3D surface without access to ground truth camera poses. To do so, we first learn a category-specific 3D template in the canonical coordinate frame together with an instance specific deformation field by leveraging synthetic 3D data. During test time, rather than directly reconstructing the shape in the camera coordinate frame, we use recent advances in point cloud canonicalization to transform a partial depth scan to the canonical space for surface reconstruction. Next, we describe first how we learn the 3D shape prior purely on synthetic data. Then we discuss how we reconstruct the surface of an observed depth image by deforming the learned canonical template shape.

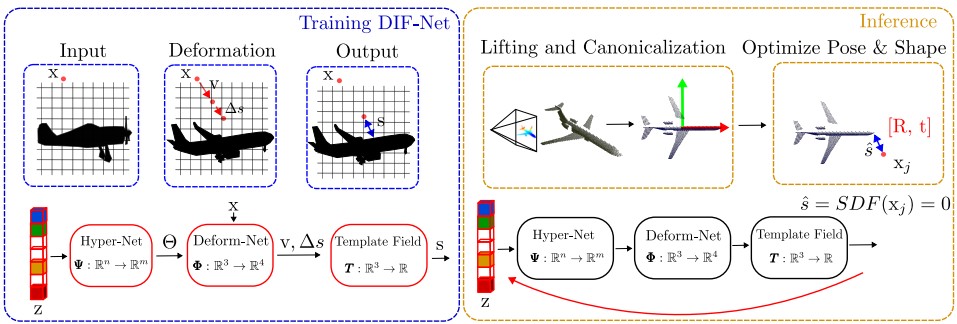

Figure 2: We train a DIF-Net as our 3D representation on purely synthetic data in an auto-decoder fashion. During inference, we lift the depth image into 3D using the known camera intrinsics and estimate an initial transformation between the camera frame and the canonical frame. We jointly optimize the object pose and 3D shape to fit the partial observation. Trainable parameters/network parts are marked in red.

## 3.1 3D SHAPE PRIOR

Given a set of 3D objects $\mathcal{O}_i$, our goal is to learn a category-specific 3D shape prior together with a latent space describing the variation in shapes. Instead of directly mapping a low dimensional latent code $z_i \in \mathbb{R}^n$ to the 3D shape, we follow recent advances in learning 3D shape priors through deformation of a canonical template (Zheng et al., 2021; Deng et al., 2021). We jointly learn our 3D shape prior, represented as a neural network, and latent codes $z_i$ through the auto-decoder framework presented in Park et al. (2019). To generate high-quality 3D reconstructions, we use signed distance fields (SDFs). SDF is a function that assigns each point $x_j \in \mathbb{R}^3$ a scalar value $s_j \in \mathbb{R}$

$$SDF(x_j) = s_j, \tag{1}$$

representing the distance to the closest object surface. The sign of $s_j$ indicates whether a point is inside (negative) or outside (positive) of the object, and the surface can be extracted as a mesh through marching cubes (Lorensen & Cline, 1987). We use a DIF-Net as our 3D shape prior network (Deng et al., 2021) and mention the necessary background for completeness. Please see Deng et al. (2021) for additional details. The 3D representation network consists of a neural template field and a deformation field. We use the template field to capture common structures among a category of shapes by keeping the weights shared across all instances in the training set. The template field takes a 3D coordinate $x_j$ as input and predicts the signed distance to the closest surface $\hat{s}$:

$$T : x_j \in \mathbb{R}^3 \to \hat{s} \in \mathbb{R} \tag{2}$$

Note that the template field does not have to represent a valid object but fuses common structures from various objects into a single neural field (see Figure 6 for some examples of the template shapes). To deform the template to a specific object instance, we use a deformation field together with a structural correction field

$$D : x_j \in \mathbb{R}^3 \to (v, \Delta s) \in \mathbb{R}^4. \tag{3}$$

The vector $v$ deforms a point in the instance space to the template space, and the correction factor $\Delta s$ modifies the SDF value of point $x_j$ if it still differs from the ground truth value. The correction factor is beneficial for categories with significant shape variations. For example, for chairs, there exist instances with and without armrests. We use a Hyper-Network (Sitzmann et al., 2019; Mitchell et al., 2019) to condition the deformation field on a latent code. With a learned template field $T$ and deformation field $D$, the SDF value of a point $x_j$ can be obtained with

$$s_j = T(x_j + v) + \Delta s = T(x_j + D_v(x_j)) + D_\Delta(x_j). \tag{4}$$

## 3.2 TRAINING DIF

During training we use the auto-decoder framework (Sitzmann et al., 2019; Park et al., 2019) to jointly learn latent codes $z_i$ and the weights of the DIF network that predicts SDF values $\hat{s} = \Psi(x)$. Given a collection of shapes with ground truth SDF values on the object surface and in free space,

we first apply an SDF regression loss from Sitzmann et al. (2020) as

$$\mathcal{L}_{sdf} = \sum_i (\sum_{x \in \Omega} |\Psi_i(x) - s| + \sum_{x \in S_i} (1 - \langle \nabla \Psi_i(x), n \rangle) + \sum_{x \in \Omega} |||\nabla \Psi_i(x)| - 1|| + \sum_{p \in \Omega \setminus S_i} \rho(\Psi_i(x))),$$
(5)

where $s$ and $n$ denote the ground truth SDF value and normal, $\nabla$ is the spatial gradient of the neural field, $\Omega$ is the 3D space in which values are sampled, and $s_j$ is the shape surface. We select an equal number of surface and free space points uniformly at random to compute this loss. The first term in Equ. 5 regresses the SDF value; the second term learns consistent normals on the shape surface, the third term is the eikonal equation that enforces unit norm or the spatial gradients, and the last term penalizes SDF values close to 0 which are far away from the object surface with $\rho(s) = \exp(-\delta \cdot |s|), \delta >> 1$. For more details on this loss, check Sitzmann et al. (2019). We further apply multiple regularization terms to help learn smooth deformations and consistent latent space. The first regularization term applies $L_2$ regularization on the embeddings as $\mathcal{L}_z = \sum_i ||z_i||_2$. Prior work by Deng et al. (2021) showed that learning a template shape that captures common attributes across a category is improved by enforcing normal consistency across all shapes by regularizing the normals of the template networks with

$$\mathcal{L}_{normal} = \sum_i \sum_{x \in S_i} (1 - \langle \nabla T(x + D_v(x)), n \rangle).$$
(6)

We further want deformations to be smooth and the optional corrections to the SDF field to be small, which is enforced with the following two loss terms $\mathcal{L}_{smooth} = \sum_i \sum_{x \in \Omega} ||\nabla D_v(x)||_2$ and $\mathcal{L}_c = \sum_i \sum_{x \in \Omega} |D_{\Delta s}(x)|$. The overall loss to training the 3D shape prior is

$$\mathcal{L} = \mathcal{L}_{sdf} + \lambda_1 L_{normal} + \lambda_2 \mathcal{L}_z + \lambda_3 \mathcal{L}_{smooth} + \lambda_3 \mathcal{L}_c,$$
(7)

with the $\lambda$ terms weighing the relative importance of each loss term.

### 3.3 POINT CLOUD LIFTING AND CANONICALIZATION DURING TESTING

Given a single RGB-D image and known camera intrinsic parameters during test time, we lift the depth image to a partial 3D point cloud. In order to predict the deformation field of the observed object, we first bring the observed partial point cloud to the canonical coordinate frame by leveraging Equi-pose (Li et al., 2021) as our pose estimation module. Equi-pose applies a SE(3)-equivariant network to learn category-specific canonical shape reconstruction and pose estimation in a self-supervised manner. By enforcing consistency between the invariant shape reconstruction and the input point cloud transformed by the estimated pose, Equi-pose can estimate the pose of the input point cloud with respect to the learned canonical frame. Therefore, we first input a complete template shape in our canonical frame to Equi-pose such that the transformation between our canonical frame and Equi-pose's canonical frame can be obtained. This way, we can transform any observed partial point cloud to our canonical frame using Equi-pose as a pose estimator. However, the estimated pose from Equi-pose can only serve as a noisy initialization. We show in Section 3.4 how our method further optimizes for the pose to achieve accurate shape reconstruction.

### 3.4 JOINTLY OPTIMIZING SHAPE AND POSE

Once we trained the 3D shape prior network and the partial input point cloud is roughly aligned in the canonical space, we reconstruct the object surface by optimizing only the latent code and the object pose with the fixed prior network. As canonical 3D reconstruction methods are sensitive to minor deviations between estimated and canonical coordinate frames 4.5 we jointly optimize the latent code $z_i$ and the initial transformation by minimizing the SDF values at the observed depth points. At the same time, we sample random points in free space for the Eikonal term to ensure that the neural field is an SDF. We represent the translation as a three-dimensional vector initialized to zero and used the continuous 6D rotation parametrization from Zhou et al. (2019) for rotations. We choose a random latent code from the learned latent space as our initialization $z_i$ and optimize

$$\min_{z_i, R, t} \mathcal{L}_{sdf} + \lambda_2 \mathcal{L}_z.$$
(8)

Note that we keep the weights of the prior network fixed to search for the latent representation that minimizes the SDF value at the points in the partial point cloud.

## 4 EXPERIMENTS

**Datasets** We include three categories in our experiments: car, chair, and airplane. We use synthetic data from the ShapeNet dataset (Chang et al., 2015) to train our deformation and template networks using 3D ground truth. Then our method trained on Shapenet is directly evaluated on the following datasets: ShapeNet (Chang et al., 2015), Pix3D chairs (Lim et al., 2013), Pascal3D+ (Xiang et al., 2014) and the DDAD (Guizilini et al., 2020) dataset. Since Pascal3D+ and Pix3D do not contain depth scans, we generate the partial point clouds by removing invisible points of the CAD models using the ground truth camera poses. Since DDAD does not provide reconstruction ground truth, we show the performance of our method on real-world noisy scans qualitatively with DDAD. See the appendix for additional information on datasets, baselines, and implementation details.

**Implementation Details** In line with prior work we train the 3D shape network on three categories in the Shapenet (Chang et al., 2015) dataset, namely *car, chair* and *plane*. The networks are trained using the Adam optimizer (Kingma & Ba, 2014). We use batch size $128$ shapes per iteration and use $4000$ points on the surface, and $4000$ randomly sampled points in free space per object. Training takes 10 hours on four NVIDIA V100 GPUs.

**Baselines** We compare against the state-of-the-art in single view, category-specific 3D object reconstruction: i) SDF-SRN (Lin et al., 2020), a neural field method that represents the object in camera coordinate frame and uses a neural renderer with silhouette supervision. ii) Closest to our method is TARS-3D Duggal & Pathak (2022), a method that uses ground truth camera poses to render a deformed template shape in canonical space to the image coordinate frame. Note that TARS-3D has access to ground truth camera poses during inference, while our method estimates this information on the fly. As our method is closely related to point cloud completion, we further compare our method against a transformer-based point cloud completion method PoinTr (Yu et al., 2021). In contrast to the baselines, our model is only trained on Shapenet and does not have access to any 3D or camera pose information during inference except the partial depth scan.

**Evaluation Metrics** In this work, we follow Tatarchenko et al. (2019) and report the F1-score at threshold $1\%$ as our primary evaluation metric. We additionally report the bidirectional chamfer distance (CD), multiplied by a factor of $1e4$ for readability.

### 4.1 3D RECONSTRUCTION ON SYNTHETIC SHAPENET DATA

Table 1 shows quantitative results of testing all approaches on the holdout test set of Shapenet. Our method outperforms the baselines in the car and plane categories with ground truth camera poses and is competitive in the chair category. We investigate cases where no ground truth camera poses are available and initialize our method and PoinTr with the Equi-pose estimates. Even without access to ground truth camera poses, our method performs comparable to or better than the baseline methods. We can see that PoinTr suffers greatly when the coordinates are not in the canonical coordinate system, showing that our approach of combining canonicalization with shape reconstruction is necessary for 3D shape reconstruction on real-world depth data. Our method's 3D reconstructions are more faithful to the underlying ground truth mesh. PoinTr outputs only a limited number of points and does not reconstruct the underlying surface, nor does it give us correspondences between shapes in a category.

Table 1: 3D reconstruction results on synthetic test data. We report chamfer distance (CD) $\downarrow$ and F-score at threshold $0.01$ (F@1%)$\uparrow$. [†] with ground truth camera poses.

| Methods | Car | | Chair | | Plane | |
|---|---|---|---|---|---|---|
| | CD ($\downarrow$) | F@1 ($\uparrow$) | CD ($\downarrow$) | F@1 ($\uparrow$) | CD ($\downarrow$) | F@1 ($\uparrow$) |
| SDF-SRN (Lin et al., 2020)[†] | 9.965 | 0.404 | 27.562 | 0.283 | 12.374 | 0.459 |
| TARS-3D (Duggal & Pathak, 2022)[†] | 10.175 | 0.412 | 28.823 | 0.272 | 11.302 | 0.418 |
| PoinTr (Yu et al., 2021)[†] | 13.249 | 0.264 | **12.834** | **0.352** | **3.637** | 0.685 |
| Ours[†] | **6.181** | **0.497** | 27.292 | 0.343 | 4.495 | **0.768** |
| PoinTr (Yu et al., 2021) | 56.435 | 0.125 | 36.714 | 0.230 | 23.713 | 0.302 |
| Ours | **9.371** | **0.439** | **32.138** | **0.334** | **16.562** | **0.631** |

## 4.2 3D RECONSTRUCTION ON SHAPENET WITH OCCLUSION

This section investigates how our method performs when the observed object is occluded. For this experiment, we generate occluded area for the Shapenet test data as presented in Figure 3. To study the influence of a wide variety of occlusions, we use rectangular overlap regions and generate overlap ratios from 5 - 85%. We apply models trained on Shapenet on the occlusion task for all methods. As shown in Table 2, our method outperforms all the baselines without access to ground truth camera poses despite not seeing any occluded data during training. Compared to the other baselines, PoinTr has a higher tolerance to occlusion. However, point cloud completion methods cannot predict the object's surface. Moreover, these methods usually predict the complete point cloud by adding predicted points to the input. Therefore, the predicted point clouds are not guaranteed to be uniformly distributed and can have a higher density around the input points, resulting in lower Chamfer distance, as shown in Figure 3. In contrast, our method does not have these drawbacks and can reconstruct the occluded surface by leveraging the learned category-level prior with high fidelity in terms of F-score.

Table 2: 3D reconstruction results on occluded data from the synthetic test set. We report chamfer distance (CD) ↓ and F-score at threshold 0.01 (F@1%)↑. [†] with ground truth camera poses.

| Methods
w. gt camera pose | Car | | Chair | | Plane | |
|---|---|---|---|---|---|---|
| | CD ($\downarrow$) | F@1 ($\uparrow$) | CD ($\downarrow$) | F@1 ($\uparrow$) | CD ($\downarrow$) | F@1 ($\uparrow$) |
| SDF-SRN (Lin et al., 2020)[†] | 25.863 | 0.243 | 156.825 | 0.122 | 72.255 | 0.256 |
| TARS-3D (Duggal & Pathak, 2022)[†] | 23.806 | 0.241 | 108.976 | 0.131 | 58.998 | 0.263 |
| PoinTr (Yu et al., 2021)[†] | 22.110 | 0.208 | **19.079** | **0.297** | **6.168** | 0.601 |
| Ours[†] | **6.457** | **0.497** | 35.972 | **0.297** | 15.725 | **0.654** |
| PoinTr (Yu et al., 2021) | 59.292 | 0.1225 | 54.327 | 0.184 | 37.425 | 0.322 |
| Ours | **11.155** | **0.390** | **45.886** | **0.273** | **22.223** | **0.580** |

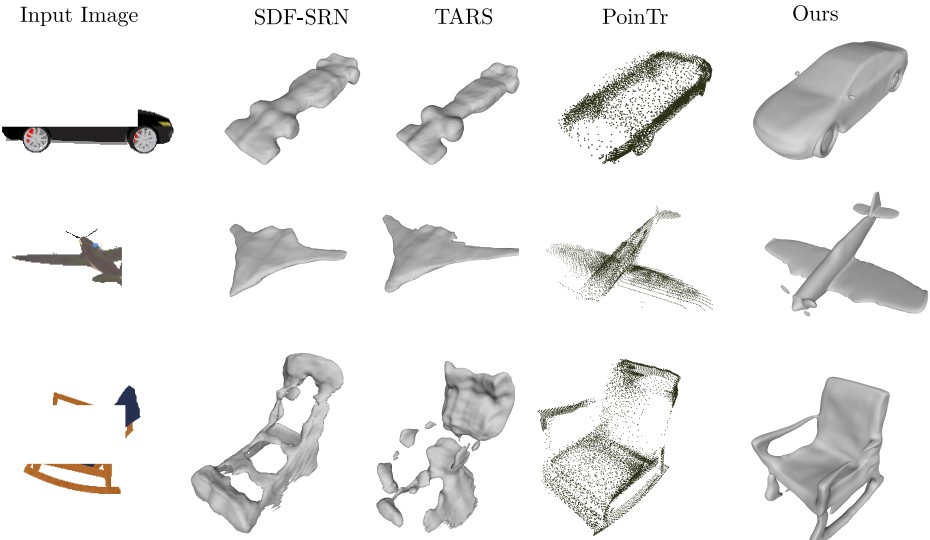

Figure 3: Qualitative result on the occluded Shapenet dataset.

## 4.3 3D RECONSTRUCTION ON PASCAL3D+ AND PIX3D

For this experiment, we test the generalization capabilities of our approach. We directly apply our model trained on Shapenet for reconstructing Pascal3D+ objects. TARS-3D and PoinTr also apply network weights trained on Shapenet to this task, while SDF-SRN is trained directly on the Pascal3D+ dataset. As shown in Table 3, our method again outperforms the other surface reconstruction methods without access to ground truth camera poses. Figure 4 shows that our method generates reasonable outputs.

Table 3: 3D reconstruction results on the Pascal3D+ and Pix3D dataset. We report chamfer distance (CD) ↓ and F-score at threshold 1% (F@1%)↑. † with ground truth camera poses.

| Methods | Pascal3D+ Car | | Chair | | Plane | | Pix3D Chair | |
|---|---|---|---|---|---|---|---|---|
| | CD (↓) | F@1 (↑) | CD (↓) | F@1 (↑) | CD (↓) | F@1 (↑) | CD (↓) | F@1 (↑) |
| SDF-SRN (Lin et al., 2020)† | 16.740 | 0.245 | 24.374 | 0.216 | 29.7457 | 0.169 | 60.432 | 0.158 |
| TARS-3D (Duggal & Pathak, 2022)† | 19.129 | 0.2427 | 79.8922 | 0.1577 | 83.866 | 0.140 | 55.555 | 0.197 |
| PoinTr (Yu et al., 2021)† | 17.859 | 0.221 | **11.769** | **0.302** | **4.701** | **0.525** | **13.092** | **0.377** |
| Ours† | **13.370** | **0.302** | 16.804 | 0.274 | 54.286 | 0.289 | 31.165 | 0.335 |
| PoinTr (Yu et al., 2021) | 80.896 | 0.064 | 21.251 | 0.216 | **35.095** | 0.231 | **35.527** | 0.283 |
| Ours | **15.516** | **0.283** | **17.328** | **0.275** | 54.849 | **0.276** | 35.729 | **0.335** |

We further test our method on another chair dataset, namely the chair category of the Pix3D dataset. As shown in Table 3, our method outperforms the baselines. PoinTr again achieves a lower Chamfer distance, which does not fully represent the reconstruction quality. As shown in Figure 4, PoinTr generates point clouds that are not uniformly distributed while our method predicts smooth surfaces.

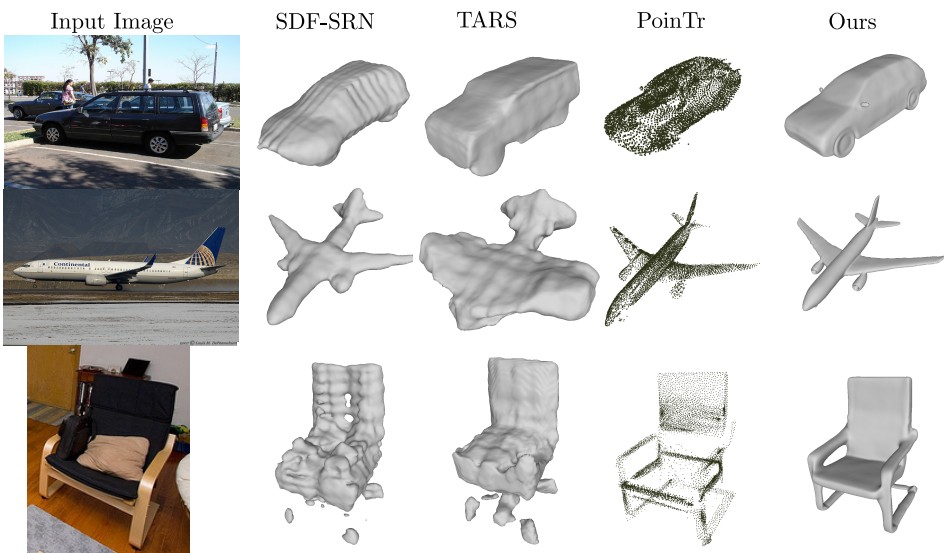

Figure 4: Qualitative results on the Pascal3D+ (top) and Pix3D (bottom) datasets.

## 4.4 3D RECONSTRUCTION ON REAL-WORLD NOISY SCANS

Finally, we apply our method trained on Shapenet directly to real-world noisy LIDAR scans. To demonstrate our tolerance to noise in the point clouds, we test our method on an autonomous driving benchmark DDAD (Guizilini et al., 2020). DDAD contains urban scenes scanned using LiDARs mounted on self-driving cars. To showcase our method, we extract frames that include other driving cars and crop the LiDAR scans of other cars with masked images. Finally, these noisy LiDAR scans are fed to the pose estimation module and our deformation field to reconstruct the surfaces. Since DDAD does not contain ground truth CAD models, we present the qualitative results in Figure 5. Note that our method does not have access to the image but only the noisy LiDAR point clouds. Despite large portions of missing parts and the noise in the LiDAR scans, our method can still reconstruct reasonable car surfaces without access to ground truth camera poses.

## 4.5 ABLATION STUDY AND FAILURE CASES

In this section, we conduct an ablation study to asses the importance of optimizing the pose during inference. The main results are shown in Table 4. On the left, we show the ground truth mesh overlaid with the partial input points with pose optimization (blue) and without (green). We can see a noticeable reduction in F1 scores and a significant reduction in the reconstruction quality. The chamfer score difference between reconstruction methods across the dataset is slight, though, confirming Tatarchenko et al. (2019) in that chamfer distance is not an ideal metric for 3D reconstruc-

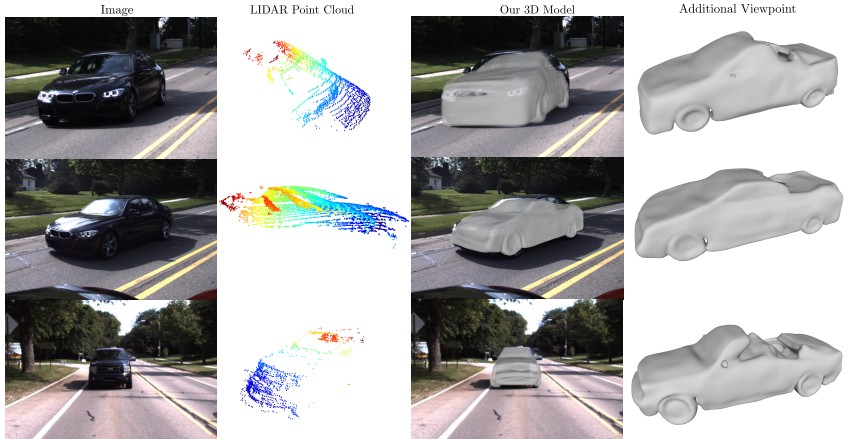

Figure 5: Qualitative results on the DDAD dataset.

tion. This ablation study shows that canonical reconstruction methods are sensitive to deviations in the estimated canonical coordinate frame. This result is also confirmed by the poor performance of PoinTr on input points where the estimated canonical coordinate frame is off (see Tables 1 and 3).

Table 4: Ablation of our method with and without pose optimization during inference. Left, we show the ground truth mesh overlayed with the optimized and non-optimized pose. We show the resulting optimized surfaces with and without pose optimization in the middle and right.

| Pose optim. | | ✓ |
|---|---|---|
| CD | 23.008 | 22.223 |
| F1 | 0.313 | 0.580 |
| Shape | | |

## 5 CONCLUSION AND FUTURE WORK

We introduced a new method for complete 3D surface reconstruction of an object from real-world depth images. Our method relies on a representation obtained solely by training on synthetic data, which allows for extracting high-quality, category-specific geometry. We showed that even small errors in pose estimation lead to significant errors in 3D reconstruction. Therefore a simple method which uses an independently trained pose estimator followed by reconstruction in the object frame does not yield good reconstruction results. Instead, we presented a finetuning scheme to optimize the object surface and pose jointly during inference. We also showed that learning strong 3D priors benefits the 3D reconstruction of occluded objects. Our method generalizes across datasets and input modalities, from dense depth images to sparse LIDAR point clouds. While our process still exhibits failure modes when the error in the estimated pose is large, this could be alleviated by combining pose estimation and 3D reconstruction in an end-to-end trainable manner. We hope our work will inspire further work in this direction.

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
