# OpenReview forum: "3D Surface Reconstruction in the Wild by Deforming Shape Priors from Synthetic Data"
_ICLR.cc/2023/Conference — Submitted to ICLR 2023_

### Official Review · Reviewer_Xiit · 2022-10-23

**Confidence:** 2
**Correctness:** 3
**Technical Novelty And Significance:** 2
**Empirical Novelty And Significance:** 2
**Recommendation:** 5

**Clarity, Quality, Novelty And Reproducibility:**

In general, the paper is well written and it is clear enough. While the contribution seems to be a bit incremental, or a combination of previous models and ideas, the experimental analysis and the final results are competitive. In terms of reproducibility, I feel the authors also provide most of the details we need to reproduce the results.

**Strength And Weaknesses:**

The authors claim their method does not require ground truth camera pose or 3D reconstruction for supervision, being the goal to learn a category-specific 3D prior model. I have no problem with that, in fact, I like the idea. However, similar ideas were exploited in the non-rigid structure-from-motion community, a community that was not included in the discussion of this paper. I know the differences, but I feel a discussion of these approaches could help the reader. We can find some works on the literature, such as: “Unsupervised 3D Reconstruction and Grouping of Rigid and Non-Rigid Categories” and “Structure from category: A generic and prior-less approach.”

The proposed method is a combination of well-known ideas and architectures in the literature. In this context, the technical contribution is a bit poor.

Be consistent with notation. For example, see x in Fig. 2 and Eqs. (1), (2), etc.

Reproducibility: some weight coefficient values, for example, see Eq. (7) are never provided in the main paper.

A calibrated camera is required. This should be commented on in the introduction.

Why not use additional categories? A priori, I think more everyday categories are included in the datasets the authors are using, so I cannot understand why just three are considered.

The authors report quantitative evaluation in terms of chamfer distance and F1 score. To be honest, for a 3D reconstruction algorithm I would hope for a different metric where we evaluate the 3D model, instead of one based on chamfer deviation.

The method provides a good performance in comparison with competing techniques. Moreover, it can handle occlusion properly. However, the occlusion seems not to be very realistic. Instead of that, I would like to see some cases where the occlusion can explain a natural and real case. The method can naturally also handle missing observations.

I agree with the authors; the proposed method produces more smooth solutions to the point of avoiding capturing details. Can the authors imagine an explanation of that?

**Summary Of The Paper:**

In this paper is proposed a method to recover (complete) the 3D reconstruction of an object category from a single depth picture. The method exploits synthetic training data and a point cloud pose canonicalization to learn a category-specific geometry. As a consequence, the learning of these 3D priors helps to retrieve the 3D reconstruction even for occluded bodies. The method is evaluated on different scenarios by using depth images or LIDAR-like point clouds; a comparison with competing techniques, ablation studies and failure cases are also reported.

**Summary Of The Review:**

As I commented before, the results obtained are promising, and even though some priors want to be simplified with respect to other techniques in the literature, the novelty of the proposal is somewhat incremental.

---

> ### Author Response · Authors · 2022-11-17
> **Rebuttal R3**
>
> Thank you for your review and suggestions that help clarify the paper. Please see our joint review regarding the novelty and contribution. We have modified the paper based on your suggestions about fixing the notations and clarify the problem setup in the Introduction.
>
> Thanks for pointing out the non-rigid structure-from-motion community. Indeed the idea of using category-level knowledge to perform pose estimation and reconstruction are similar. We did not include it in our related work as they do not focus on surface reconstruction of rigid objects.
>
> Reproducibility: The weights of Eq. 7 are mentioned in the supplementary material.
> Occlusions: We generate the occluded dataset with rectangular patches to cover 5 - 85% occlusion. Although using rectangular patches is not realistic, it does make the computation of overlap ratios easier while still providing very challenging reconstruction scenarios. This experiment is intended for analyzing quantitatively how our method performs under different levels of occlusion.
>
> Additional categories: We agree with the reviewer that adding additional categories would help evaluate our method more thoroughly. The three categories that we included in the paper are the most common categories among SOTA category-specific 3D reconstruction methods.
> A different metric other than Chamfer: We agree with the reviewer that Chamfer distance does not capture the reconstruction quality well, which we also verified in Table 4. Therefore, we also include the F1 metric, which is a widely used metric among the literature. Both of these metrics are widely used to evaluate the task. Other than F1 score, does the reviewer recommend other metrics that we should include?
>
> Smooth solutions: In our method, we optimize for the latent shape code and the camera pose such that the estimated surface best fits the observed points. Since the latent space is trained with perfect shapes, the optimized test shape that best fits the observed partial point cloud will also represent a smoother surface even though the observation contains noise.

---

### Official Review · Reviewer_qugy · 2022-10-25

**Confidence:** 5
**Correctness:** 4
**Technical Novelty And Significance:** 2
**Empirical Novelty And Significance:** 2
**Recommendation:** 5

**Clarity, Quality, Novelty And Reproducibility:**

The idea has been clearly presented. However, since the method consists of several steps and some tech details are not provided, the work might be hard to reproduce.

**Strength And Weaknesses:**

Strength:
- The method achieves better results compared to some recently proposed learning based methods. It also addresses some of their limitations which moves it closer to practical application (still a long way to go though)
- The method shows good results on the picked occlusion cases. It would be interesting to see a more thorough evaluation or demonstration.

Weakness:
- It is not clear how fast the method runs in practice, especially considering that it requires segmenting the objects from the image or point cloud and lifting the point cloud to the canonical coordinate frame.
- The proposed method seems to be a combination of ideas from existing works. The novelties over these existing works are not clearly discussed.
- Several parts of the paper need to be revised.
  - Abstract "A limitation of current color image-based 3D reconstruction models is that they do not generalize across datasets": this should only apply to learning based methods.
  - 2.1 "Neural rendering and neural fields provide an alternative representation to overcome these limitations. Surface fields, such as Signed Distance Functions (SDFs) (Xu et al., 2019; Lin et al., 2020; Duggal & Pathak, 2022) or volumetric representations such as occupancy (Ye et al., 2021)": both SDF and occupancy grid are classic represetnations, they are not proposed or provided by neural rendering and neural fields.
  - 3.1 "Instead of directly mapping a low dimensional latent code zi ∈ Rn to the 3D shape through concatenation": concatenation is quite confusing.
  - Eq 2: what is \hat{s}?
  - 3.1 What does "instance space to the template space" mean?

**Summary Of The Paper:**

This paper present a deep learning based pipeline to reconstruct the 3D surface of an object from a RGBD image. A network is trained to encode the category shape priors and then deform the shape priors to fit the object point cloud. A separate network is needed to transorm the object point cloud from the camera coordinate to the canonical coordinate frame. The proposed method is evaluated on both synthetic and real datasets and shows superior performance to the compared peer works.

**Summary Of The Review:**

The work seems to be a cobination of existing ideas and the novelty is marginal. My current rating is borderline reject.

---

> ### Author Response · Authors · 2022-11-17
> **Rebuttal R2**
>
> Thank you once again for your review and suggestions to clarify the paper. For our reply regarding the novelty in the paper, please see our joint reply. We have also added all your suggestions to the paper. Please see our comments regarding your questions below.
>
> “It is not clear how fast the method runs in practice”: The part of our method that takes the longest is the pose optimization at test time. While this is a function of the number of iterations, in our tests, we use 30 iterations at test time, which takes roughly 4 seconds on a single V100 GPU. We have added this to the supplementary material in Section A2.
>
> Reproducibility: We will release our code, data, and network weights along with the final version of the paper.
>
> Occlusions: We generate various occlusions, ranging from 5 to 85% overlap, using rectangular patches. We use rectangular patches because they simplify the computation of overlap ratios. We have added additional details in the experiment section.

---

### Official Review · Reviewer_aneS · 2022-10-25

**Confidence:** 3
**Correctness:** 2
**Technical Novelty And Significance:** 1
**Empirical Novelty And Significance:** 1
**Recommendation:** 3

**Clarity, Quality, Novelty And Reproducibility:**

The presentation of the paper is not very clear. Quality, novelty and reproducibility are also not very convincing.

**Strength And Weaknesses:**

### Strength
- Qualitative and quantitative results seem impressive, there are notable improvements over previous work.
- The appendix is useful to understand more details of the work.

 ### Weakness
- Overall the work feels more like a pipeline to combine different components together. It is not fully clear to me what the major technical contribution of the paper is, compared to the previous work. It would be useful if authors can highlight and contrast over the previous work.
- The writing and presentation of the paper can be improved. Currently it's not very easy to follow the paper.
- Ablation study and analysis is a bit lacking, thus it's also hard to understand the individual impact of the components.

### Additional Qs:
- In table 2, why the improvement of car is larger compared to other categories?


**Summary Of The Paper:**

The paper presents a method for single-view category-specific 3D reconstruction. The pipeline starts with deriving a canonical pose with a partial point cloud. Then a neural deformation is used to reconstruct the object's 3D surface. Finally a joint optimization of pose and shape is to further improve the results.


**Summary Of The Review:**

To me the paper is clearly below the bar from multiple aspects: clarity of presentation, technical novelty and insufficient experimental validation. Thus I would vote for reject.

---

> ### Author Response · Authors · 2022-11-17
> **Rebuttal R1**
>
> Thank you once again for your review. For our reply regarding the novelty in the paper, please see our joint reply.
>
> Regarding ablations, we mainly wanted to show that a vanilla combination of components suffers from misalignments in the estimated object pose. What additional ablations could we show to improve the paper?
> Regarding the presentation, we have adapted the other reviewers' suggestions to make the presentation clearer.
>
> In table 2, why the improvement of car is larger compared to other categories? Compared to airplanes and chairs, the car category has less shape variety and more symmetry; therefore, morphing the car template to an observed car can require less deformation and achieve a better quantitative result.

---

### Author Response · Authors · 2022-11-17
**Rebutal**

We thank the reviewers for their insightful comments and suggestions to strengthen our paper. We want to highlight that the reviewers found the “qualitative and quantitative results impressive” [R1], “the method achieves better results compared to recently proposed learning based approaches”, shows “good results on the picked occlusion cases” [R2], and “The method provides a good performance in comparison with competing techniques” [R3].

The main weakness of the paper highlighted by the reviewers is the perceived lack of technical novelty. We disagree with this remark, as we believe that novelty is creating insight which will benefit the research community. Our paper is the first to show that combining canonicalization and surface reconstruction can achieve high performance for category-specific 3D reconstruction and we believe that these insights will compel other research in a similar direction.
Existing category-specific single-view 3D reconstruction methods (Lin et al., 2020; Duggal & Pathak 2022) assume known perfect camera pose during test time. However, acquiring ground truth camera poses for in-the-wild images is non-trivial. Our presented optimization method performs not only 3D reconstruction but also handles noisy camera poses, which can come from an off-the-shelf pose estimation module. Point cloud completion-based methods also assume known camera poses during test time. We show how these methods suffer from noisy camera poses in the experiment section. In contrast, our method is less sensitive to the noise in the camera pose and can predict smooth surfaces. Although the neural deformation field is a well-known method for the shape morphing of “complete”  shapes, it has yet to be applied to partial observations. The original DIF-Net (Deng et al., 2021) works with only complete, perfectly aligned shapes. Our proposed pipeline extends the method to a more challenging setup where we are given unposed partial observations, which also suffer from noise and occlusion.
We would like to reiterate that  in contrast to existing work, our contributions are: 1. We demonstrate better generalizability by training solely on synthetic datasets and testing different real-world datasets. 2. Our method works on a more challenging real-world setup that has not been addressed thus far. We investigate in-the-wild images where the camera pose is unknown, and the partial observation can contain noisy and occluded objects.

In this work, we present the first prototype to show the usefulness of canonicalization for shape reconstruction and intend to extend this work in the future.
We thank the reviewers again and continue with specific comments of individual reviewers.

---

### Decision · Program_Chairs · 2023-01-20

**Decision:**

Reject

**Justification For Why Not Higher Score:**

There are significant concerns regarding the empirical results/setup (i.e. comparing an approach with RGB-D input to prior RBG-based methods) as well contributions (while the pipeline is novel, it primarily combines existing components).

**Justification For Why Not Lower Score:**

N/A

**Metareview: Summary, Strengths And Weaknesses:**

The work tackles the task of 3D reconstruction given a single RGB-D image. The reviewers raise concerns regarding the novelty of the method as, although the overall pipeline is interesting, the core components (shape completion, canonicalization) are built on existing approaches. Further, the AC is also concerned that the empirical results compare this approach to baselines which do not require depth and this is not a fair comparison. Moreover, the text claims that these RGB methods require camera pose an inference but this maybe incorrect e.g. Sec 4. of TARS states “Each training example consists of cropped RGB image (centered around the object), corresponding segmentation map and camera pose. At inference time, we only need the object image as input.” Overall, given the concerns in the empirical results and contributions, the AC is leaning towards rejection.